# Co-Circulation of Two Independent Clades and Persistence of CHIKV-ECSA Genotype during Epidemic Waves in Rio de Janeiro, Southeast Brazil

**DOI:** 10.3390/pathogens9120984

**Published:** 2020-11-26

**Authors:** Allison Araújo Fabri, Cintia Damasceno dos Santos Rodrigues, Carolina Cardoso dos Santos, Flávia Löwen Levy Chalhoub, Simone Alves Sampaio, Nieli Rodrigues da Costa Faria, Maria Celeste Torres, Vagner Fonseca, Patricia Brasil, Guilherme Calvet, Luiz Carlos Junior Alcantara, Ana Maria Bispo de Filippis, Marta Giovanetti, Fernanda de Bruycker-Nogueira

**Affiliations:** 1Flavivirus Laboratory, Oswaldo Cruz Institute, FIOCRUZ, Rio de Janeiro 21040-360, Brazil; allison.fabri@ioc.fiocruz.br (A.A.F.); cintia.damasceno@ioc.fiocruz.br (C.D.d.S.R.); carolina.santos@ioc.fiocruz.br (C.C.d.S.); flavia.levy@ioc.fiocruz.br (F.L.L.C.); sampaiosa@ioc.fiocruz.br (S.A.S.); nielircf@ioc.fiocruz.br (N.R.d.C.F.); maria.torres@ioc.fiocruz.br (M.C.T.); luiz.alcantara@ioc.fiocruz.br (L.C.J.A.); ana.bispo@ioc.fiocruz.br (A.M.B.d.F.); 2KwaZulu-Natal Research Innovation and Sequencing Platform (KRISP), School of Laboratory Medicine and Medical Sciences, Nelson R Mandela School of Medicine, College of Health Sciences, University of KwaZulu-Natal, Durban 4001, South Africa; 217082126@stu.ukzn.ac.za; 3Laboratório de Genética Celular e Molecular, Instituto de Ciências Biológicas, Universidade Federal de Minas Gerais, Belo Horizonte 31270-901, Brazil; 4Coordenação Geral dos Laboratórios de Saúde Pública/Secretaria de Vigilância em Saúde, Ministério da Saúde, (CGLAB/SVS-MS) Brasília, Distrito Federal 70719-040, Brazil; 5Evandro Chagas National Institute of Infectious Diseases, FIOCRUZ, Rio de Janeiro 21.040-900, Brazil; patricia.brasil@ini.fiocruz.br (P.B.); guilherme.calvet@ini.fiocruz.br (G.C.)

**Keywords:** chikungunya virus, ECSA genotype, persistence, co-circulation, rio de janeiro, Brazil

## Abstract

The Chikungunya virus infection in Brazil has raised several concerns due to the rapid dissemination of the virus and its association with several clinical complications. Nevertheless, there is limited information about the genomic epidemiology of CHIKV circulating in Brazil from surveillance studies. Thus, to better understand its dispersion dynamics in Rio de Janeiro (RJ), one of the most affected states during the 2016–2019 epidemic waves, we generated 23 near-complete genomes of CHIKV isolates from two main cities located in the metropolitan mesoregion, obtained directly from clinical samples. Our phylogenetic reconstructions suggest the 2019-CHIKV-ECSA epidemic in RJ state was characterized by the co-circulation of multiple clade (clade A and B), highlighting that two independent introduction events of CHIKV-ECSA into RJ state have occurred between 2016–2019, both mediated from the northeastern region. Interestingly, we identified that the two-clade displaying eighteen characteristic amino acids changes among structural and non-structural proteins. Our findings reinforce that genomic data can provide information about virus genetic diversity and transmission dynamics, which might assist in the arbovirus epidemics establishing of an effective surveillance framework.

## 1. Introduction

Chikungunya virus (CHIKV) is an RNA alphavirus belonging to the *Togaviridae* family, transmitted by *Aedes aegypti* and by *A. albopictus* mosquitoes [1,2]. Infection with CHIKV typically causes a self-limiting febrile illness, the chikungunya fever, and common clinical manifestations of the disease include fever, muscle pain, rash, and severe joint pain, which may last for months to years [1]. Chikungunya fever appears also to be linked with long-lasting rheumatic disorders presenting as acute and chronic polyarthralgia/polyarthritis, which lead to functional impairment affecting daily living activities up to several years after infection [1,2].

The first identified outbreak of chikungunya was reported in 1952 in Tanzania, East Africa, and since then it has been responsible for important emerging and re-emerging epidemics in several tropical and temperate regions [3,4].

Four distinct CHIKV genotypes (or lineages), have been already identified and named based on their geographical distribution: (i) the West African; (ii) the East/Central/South African (ECSA); (iii) the Asian; (iv) the Indian Ocean Lineage (IOL), which emerged from the ECSA lineage between 2005 and 2006 [5,6]. In the Americas, the first autochthonous CHIKV transmission was reported in 2013, being characterized by the circulation of the Asian genotype [7]. A year later, in 2014, in Brazil, the co-circulation of the Asian and ECSA genotypes were reported in Northern and Northeastern, respectively [8,9]. Since then, the ECSA genotype has been detected in several other Brazilian states in the northeastern, southeastern, and northern regions, exerting a serious threat to public health [10,11,12,13,14,15,16,17].

CHIKV infections in Brazil accounted for 132,205 probable cases in 2019, with the southeastern region responsible for approximately more than 92,000 cases [18]. In 2019, Rio de Janeiro (RJ) state reported 86,264 CHIKV suspected cases until epidemiological week 52, which is approximately 65% of all the probable cases notified in the country [18].

The RJ, located in southeast Brazil, is the second-most populous state in the country and has a territorial area of 43,750,427 km^2^, with an estimated population of 17,264,943 (demographic density: 365.23 hab/km^2^) [19]. It is considered an important economic and an important tourist destination, geographically is divided into six mesoregions: Northwest Fluminense, North Fluminense, Center Fluminense, *Baixadas*, South Fluminense, and the metropolitan region [20]. Historically, it has also been described as an important gateway of several mosquito-borne viruses, including dengue, yellow fever, and zika viruses.

The first cases of autochthonous transmissions of CHIKV in Rio de Janeiro were reported in 2015, indicating the establishment of the circulation of the ECSA [10,13]. Nevertheless, the shortage of complete genome sequences available impairs our understanding regarding its dynamic dispersion into the state. Thus, in this study, we generate 23 new CHIKV near-complete genome sequences from the 2019 epidemic sampled in two different cities located in the Metropolitan mesoregion of the Rio de Janeiro state (Rio de Janeiro and Duca de Caixia cities), with the aim of providing an overview of the circulation and dispersion events of the virus in that state.

## 2. Results

To better understand the 2019 CHIKV epidemic in some of the most affected municipalities in Rio de Janeiro, we generated 23 CHIKV near-complete genomes (coverage range 77.3–93.8%, mean = 91.2%) from serum samples using a nanopore sequencing approach. PCR cycle threshold (Ct) values were on average 1307 (range: 5.75 to 20.89) (Table 1). Most of the isolates (*n* = 17) belonged to patients that reside in the municipality of Rio de Janeiro, the capital of the RJ state, located in the metropolitan region. The remaining six samples were from a different neighboring municipality, the Duque de Caxias municipality, also belonging to the same metropolitan region of the state.

Of the 23 samples, 19 were from adult patients (>18 years), 1 from an infant (1 year), and 3 from newborns (1 and 7 days; 1 month). These samples were from 11 female (9 adults and 2 newborns) and 12 male (10 adults, 1 infant and 1 newborn). None of the patients have reported travels to other previous epidemic areas, as indicated by epidemiological data obtained from the local surveillance service. Sequencing statistics and epidemiological details of the sequences generated here are available in Table 1.

### 2.1. Phylogenetic Analyses

To investigate the phylogenetic relationships of the 2019-CHIKV strains circulating in Rio de Janeiro state, we estimated a preliminary Maximum Likelihood (ML) phylogeny from a dataset containing 767 reference sequences from the four genotypes plus the newly 23 sequences generated in this study (*n* = 790 sequences) (Appendix A). Our ML phylogeny revealed that the newly generated CHIKV sequences belong to the ECSA genotype and cluster together with other Brazilian strains belonging to the same genotype (Appendix A). These results were also confirmed by using the phylogenetic arbovirus genotyping tool (https://www.genomedetective.com/).

Further, in order to investigate the Brazilian ECSA clade in more detail, we built a second dataset including all ECSA taxa from Brazil (ECSA-BR dataset, *n* = 96), and we performed a Bayesian molecular clock reconstruction. A regression of genetic divergence from root to tip against sampling dates confirmed sufficient temporal signal (R^2^ = 0.75). Time-measured phylogenetic analysis reveals that the novel isolates were organized into two distinct clades, named hereafter as clades A and B.

Of the 23 samples, 21 were grouped in a well-supported monophyletic clade (posterior probability = 1) containing other sequences from the RJ state collected between 2016 and 2017 (clade A), suggesting that since its first introduction in the state, the virus persisted during at least during a 3-year period. The time to the most recent common ancestor (tMRCA) of this clade was estimated to be April 2017 (95% HPD: January to October 2017).

Interestingly, the other two samples grouped within a distinct clade (clade B), which also includes another sample from the RJ state isolated in 2018 (posterior probability = 1) (Figure 1) which indicate a more recent re-introduction of the CHIKV-ECSA genotype into RJ state dated back to July 2018 (95% HPD: January to October 2018).

Together, our results suggest that since 2015, Rio de Janeiro has experienced two independent events of CHIKV-ECSA introduction, both mediated by northeastern region where this lineage was first detected in late 2014.

### 2.2. Molecular Characterization of Newly CHIKV Sequences from Rio de Janeiro State

The presence of amino acid substitutions was investigated in the 23 newly near-complete genomes obtained in this study in comparison with the reference genome strain isolated in Feira de Santana, Bahia state in Mid-2014 (Accession Number: KP164568). We found evidence of synonymous and non-synonymous amino acid (aa) changes among the two clades (clade A and B), in the non-structural and structural proteins.

In more depth, we identified a total of 8 amino acid substitutions among the NSP2, NSP3, and NSP4 genes, and 10 among the E2, 6k, and E1 structural genes (Table 2).

Among clade A we identified three conserved aa positions in the NSP2 (P352A; A545S) and E1 (K211T) proteins that have also been identified in all the other strains that are clustering together in the same clade, sampled in Rio de Janeiro between 2016 and 2017.

Moreover, we also found that our newly isolates presented aa changes in the E1 (V269M; A305T) protein that appears to be characteristic of the 2019-CHIKV-ECSA epidemic (Table 2).

For the Clade B, we identified aa change signature in our 2019 strains located in the NSP2 (A57V) and E2 (R178H) proteins, respectively (Table 2).

Beside those aa changes among the two clades we also identified a total of six non-synonymous aa changes: one in NSP2 (A545S, non-polar to neutral polar), one in E2 (T74M, polar to non-polar), and four in E1 (A98T, non-polar to neutral polar neutral; D151V, acid polar to non-polar; K211T, basic polar to neutral polar; A305T, non-polar to neutral polar) proteins.

## 3. Discussion

Since CHIKV was first detected in Brazil in 2014 [8,9] more than 780,000 cases have been notified [18,21,22,23,24]. Despite its synchronicity, there is still limited information about the genomic epidemiology of CHIKV during the Brazilian epidemics. To provide more information about the dynamics of CHIKV epidemics in Brazil, we generated 23 new near-complete genome sequences from Rio de Janeiro that had the first detection of CHIKV in 2016 and experienced an explosive outbreak in 2019 [10,12,13,14].

Our results revealed that the 2019-CHIKV-ECSA epidemic in RJ state was characterized by the co-circulation of two clades (clade A and B), in line with previous findings [13] highlighting that these two different clades are persistent and responsible for the latest epidemics registered in the state. Time-scale phylogenetic analysis estimated the tMRCA of the Clade A and B to be April 2017 (95% HPD: January to October 2017), and July 2018 (95% HPD: January to October 2018), respectively.

Together, our results reinforce that two independent introduction events of CHIKV-ECSA have occurred between 2016–2019 into Rio de Janeiro, both mediated from the Northeastern region, which has played an important role in the introduction and establishment of the ECSA genotype into the country. Both strains’ introduction and their persistence along time depict a complex dynamic transmission between the epidemic seasons and sampled locations. Several factors, including the high population density, low-income level, and precarious sanitary conditions, high air connectivity, and land transport between different regions of the country, might be contributing to the modeling of this complex viral spread scenario between the different Brazilian regions.

Analysis of the newly 23 isolates allowed us to display characteristic nucleotide (nn) signatures responsible of conservative and semi-conservative amino acid (aa) changes among the structural and non-structural proteins between the two clades, some of which appear to be characteristic of the 2019-CHIKV-ECSA epidemic.

Although most of the substitutions identified appear to be synonymous, it is important to note that among the non-synonymous ones, five have been identified in the Envelope protein. This protein presents the high antigenic variability and it is considered an important viral structure for host-cells infection, since, through it, the virus attaches to the cells. Additionally, it has also been suggested that this protein plays an important role during the viral replication [25,26]. These changes in the amino acid composition in of this region may be relevant and require further investigation and potential surveillance during epidemics seasons.

Furthermore, we did not detect the A226V nor K211E aa substitutions (residues located in the E1 protein) among the samples under study. These mutations were previously described as responsible to increase the CHIKV transmission in *Ae. albopictus* and *Ae. aegypti* mosquitoes, respectively [27,28].

Our study shows that genomic surveillance strategies, as previously suggested [29,30,31] might play an essential role in monitoring the spread as well as the diversity of emerging and re-emerging mosquito-borne viruses, which is fundamental to assist public health policies.

## 4. Materials and Methods

### 4.1. Ethical Statement

The strains analyzed in this study belong to a previously gathered collection from the Flavivirus Laboratory, IOC/FIOCRUZ, Rio de Janeiro, Brazil, obtained from human serum from an ongoing Project reviewed and approved by local Ethics Committee CAAE: 90249218.6.1001.5248 from the Oswaldo Cruz Foundation. The informed consent was obtained from all subjects. Samples were chosen anonymously, based on the laboratorial results and clinical manifestations available on the Laboratory database. All methods were performed in accordance with relevant guidelines and regulations.

### 4.2. Sample Collection and RT-qPCR Diagnosis

Serum samples (*n* = 23) from Chikunungya suspected patients were screened for CHIKV RNA detection in the Regional Reference Laboratory of Flavivirus (LABFLA) at the Oswaldo Cruz Foundation. Samples were obtained from 0 to 11 days after the onset of symptoms.

Viral nucleic acid extraction was performed using the Magmax Pathogen RNA/DNA kit (Thermo Fisher Scientific, Waltham, MA, USA) and the KingFisher Flex Purification System (Thermo Fisher Scientific, Waltham, MA, USA) according to the manufacturer’s instructions. The Molecular diagnostic assay was performed by Real-Time RT-PCR using a molecular ZDC kit (Bio-Manguinhos, Rio de Janeiro, Brazil) produced by Bio-Manguinhos, FIOCRUZ, on an Applied Biosystems 7500 Real-Time PCR System machine (Thermo Fisher Scientific, Waltham, MA, USA). All procedures were conducted in biological safety cabinets located in physically separated areas. Negative controls were used in all reactions.

### 4.3. Synthesis of cDNA and Multiplex Tiling PCR

DNA amplification and sequencing were attempted on the 23 selected RT-PCR positive samples that exhibited Ct-values <38, in order to increase the viral genome coverage by nanopore sequencing (selection based on DNA concentration after clean-up being >4 ng/μL) [32].

Extracted RNA was converted to cDNA using the Protoscript II First Strand cDNA synthesis Kit (New England Biolabs, Hitchin, UK) and random hexamer priming [32]. Then, a multiplex tiling PCR was conducted using Q5 High Fidelity Hot-Start DNA Polymerase (New England Biolabs, Hitchin, UK) and CHIKV sequencing primers scheme (primers are divided into two separated pools, A and B) designed by Quick and collaborators using Primal Scheme (http://primal.zibraproject.org) [32]. The thermocycling and reaction conditions were previously reported in [32].

### 4.4. Library Preparation and Nanopore Sequencing

Amplicons were purified using 1× AMPure XP Beads (Beckman Coulter, Brea, CA, USA) and cleaned-up PCR products concentrations were measured using Qubit dsDNA HS Assay Kit (Thermo Fisher Scientific, Waltham, MA, USA). on a Qubit 3.0 fluorimeter (Thermo Fisher Scientific, Waltham, MA, USA). DNA library preparation was performed using the Ligation Sequencing Kit (Oxford Nanopore Technologies, Oxford, UK) and Native Barcoding Expansion 1–24 kit (Oxford Nanopore Technologies, Oxford, UK), whose reactions conditions have already been described [32], with the following modifications: The same sample was added to both sequencing primers pools (A and B, separated tubes) during multiplex tiling PCR. After PCR, each pool was purified, and its DNA concentration was quantified using Qubit. Then, both pools (A and B) were mixed in a single tube (taking in consideration the DNA concentrations of each pool), and one barcode was used per sample in order to maximize the number of samples per flow cell. Sequencing library was generated from the barcoded products using the Genomic DNA Sequencing Kit SQK-MAP007/SQK-LSK208 (Oxford Nanopore Technologies, Oxford, UK). Sequencing library was loaded onto a R9.4 flow cell (Oxford Nanopore Technologies, Oxford, UK). Sequencing was performed for 3 h on MinION device (Oxford Nanopore Technologies, Oxford, UK). Reads were basecalled using Guppy and barcode demultiplexing was performed using qcat. Consensus sequences were generated by de novo assembling using Genome Detective (Available at: https://www.genomedetective.com) [33].

New genome sequences obtained in this study have been deposited in GenBank under accession numbers MT933029 to MT933051.

### 4.5. Phylogenetic and Bayesian Analysis

The 23 new genomic sequences reported in this study were initially submitted to a genotyping analysis performed by Genome Detective virus tool online (https://www.genomedetective.com/). New sequences were aligned to 767 complete or almost complete CHIKV genomic sequences (>10,000 bp), retrieved from VIPR Virus Pathogen Resource (https://www.viprbrc.org) in February 2020 and covering all four existing genotypes. Alignment was performed using MAFFT online program [34]. The sequences’ analysis, edition, and molecular characterization were performed using the Bioedit (http://www.mbio.ncsu.edu/bioedit/bioedit.html). The complete dataset was assessed for presence of phylogenetic signal by applying the likelihood mapping analysis implemented in the IQ-TREE 1.6.8 software (http://www.iqtree.org) [35]. A maximum likelihood (ML) phylogeny was reconstructed from the dataset (*n* = 790) using IQ-TREE 1.6.8 software under the GTR+G+I nucleotide substitution model with four gamma categories, which was inferred in jModelTest (https://github.com/ddarriba/jmodeltest2) as the best fitting model [36]. GenBank accession numbers, countries of origin and year of isolation of all included sequences are shown in Appendix A.

From the generated maximum likelihood (ML) phylogeny using the concatenated dataset we selected all ECSA taxa from Brazil (*n* = 96). In order to investigate the temporal signal in our CHIKV-ECSA dataset, we regressed root-to-tip genetic distances from this ML tree against sample collection dates using TempEst v 1.5.1 (http://tree.bio.ed.ac.uk) [37].

The ML phylogeny was used as a starting tree for Bayesian time-scaled phylogenetic analysis using BEAST 1.10.4 (http://beast.community/index.html) [38]. We employed a stringent model selection analysis using both path-sampling (PS) and stepping stone (SS) procedures to estimate the most appropriate molecular clock model for the Bayesian phylogenetic analysis [39]. Were tested: (a) the strict molecular clock model, which assumes a single rate across all phylogeny branches, and (b) the more flexible uncorrelated relaxed molecular clock model with a lognormal rate distribution (UCLN) [40]. Both SS and PS estimators indicated the uncorrelated relaxed molecular clock as the best fitted model to the dataset under analysis. Besides, we have used the HKY+G4 codon partitioned (CP)1+2,3 substitution model and the Bayesian SkyGrid coalescent model of population size and growth [40,41]. We computed MCMC (Markov chain Monte Carlo) duplicate runs of 100 million states each, sampling every 10,000 steps for the ECSA-BR dataset. Convergence of MCMC chains was checked using Tracer v.1.7.1 [42]. Maximum clade trees were summarized from the MCMC samples using TreeAnnotator (http://beast.community/index.html) after discarding 10% as burn-in.

### 4.6. Epidemiological Data Assembly

Data of monthly notified CHIKV cases were supplied by the Health Surveillance System of the Rio de Janeiro state and were plotted using the R software version 3.5.1.

## Figures and Tables

**Figure 1 pathogens-09-00984-f001:**
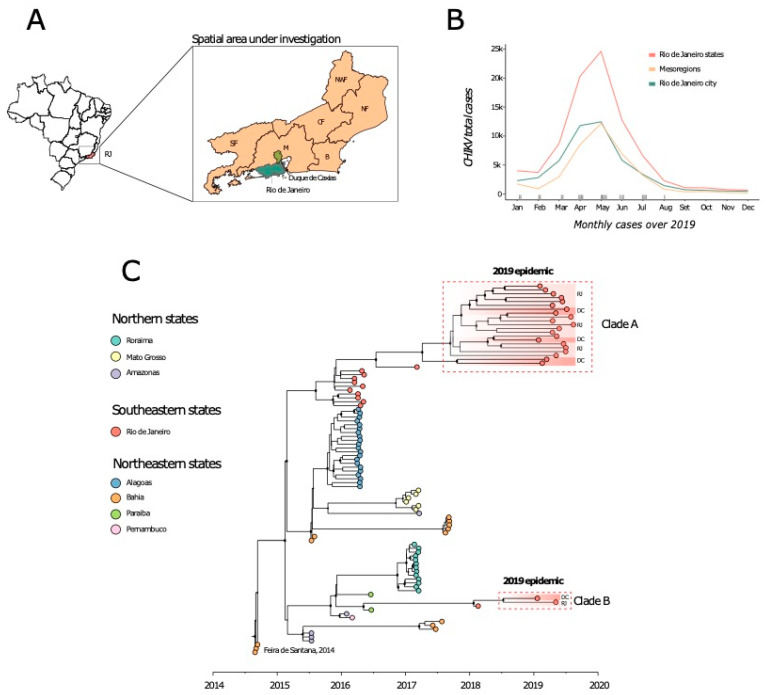
Investigation of the CHIKV-ECSA genotype in Rio de Janeiro state, southeast Brazil. (**A**) Brazilian map showing the spatial area under investigation. The municipalities where the sequences were sampled have been highlighted in green (Rio de Janeiro) and light green (Duque de Caxias). Letters in the map indicate the 6 Rio de Janeiro mesoregions: NFW = Northwest Fluminense; NF = North Fluminense; CF = Center Fluminense; B = Baixadas; SF = South Fluminense; M = metropolitan region. (**B**) Number of CHIKV probable cases reported during the 2019 epidemic in Rio de Janeiro state. (**C**) Maximum clade credibility tree (MCCT) obtained using the 23 newly CHIKV near-complete genome sequences plus 73 publicly available Brazilian CHIKV-ECSA sequences. Circles along branches represent clade posterior probability >0.90. Colors represent different locations.

**Table 1 pathogens-09-00984-t001:** Epidemiological data and sequencing statistics for the sequenced samples.

ID	Sample	Collection Date	Days of Symptoms	Sex	Age	District	City	Ct	Coverage (%)	Reads	Accession Number
CHIKV-1	Serum	2019-01-21	4	F	28	Vila Maria Helena	Duque de Caxias	20.89	92.5	86,362	MT933029
CHIKV-2	Serum	2019-01-28	5	F	26	Piabetá	Duque de Caxias	20.74	92.7	62,250	MT933030
CHIKV-3	Serum	2019-02-18	2	M	66	Parque Senhor do Bonfim	Duque de Caxias	19.15	77.3	105,226	MT933031
CHIKV-4	Serum	2019-02-05	0	NB-M	1 m	Madureira	Rio de Janeiro	17,71	92.8	25,250	MT933032
CHIKV-5	Serum	2019-03-08	1	F	35	Senador Vasconcelos	Rio de Janeiro	17.91	92.8	29,490	MT933033
CHIKV-6	Serum	2019-03-15	5	M	1	Parque Lafaiete	Duque de Caxias	9.06/9.13	93.6	67,535	MT933034
CHIKV-7	Serum	2019-04-17	2	F	71	Quintino Bocaiúva	Rio de Janeiro	10.77/11.17	92.7	43,062	MT933035
CHIKV-8	Serum	2019-04-17	2	M	61	Manguinhos	Rio de Janeiro	9.67/12.74	93.6	72,337	MT933036
CHIKV-9	Serum	2019-04-18	2	M	58	Parque Anchieta	Rio de Janeiro	13.67/13.68	85.6	133,675	MT933037
CHIKV-10	Serum	2019-04-22	1	M	88	Engenho de Dentro	Rio de Janeiro	9.48/11.46	93.6	50,757	MT933038
CHIKV-11	Serum	2019-05-06	4	M	77	Oswaldo Cruz	Rio de Janeiro	12.05/12.47	93.7	63,659	MT933039
CHIKV-12	Serum	2019-05-06	2	F	59	Nossa Senhora do Carmo	Duque de Caxias	12.76/12.76	84.9	133,090	MT933040
CHIKV-13	Serum	2019-05-08	2	M	47	Jacarepaguá	Rio de Janeiro	13.5/13.5	92.8	63,092	MT933041
CHIKV-14	Serum	2019-05-11	1	M	35	NI	Rio de Janeiro	12.31/15.98	90.9	113,375	MT933042
CHIKV-15	Serum	2019-06-07	2	M	42	NI	Rio de Janeiro	13.47/13.53	92	81,734	MT933043
CHIKV-16	Serum	2019-06-13	1	NB-F	1d	São Cristóvão	Rio de Janeiro	8.09/7.57	93.5	72,843	MT933044
CHIKV-17	Serum	2019-06-19	1	M	57	Engenho de Dentro	Rio de Janeiro	12.97/10.52	92.7	77,390	MT933045
CHIKV-18	Serum	2019-07-01	2	F	50	Bonsucesso	Rio de Janeiro	13.8/15.2	93.5	69,270	MT933046
CHIKV-19	Serum	2019-07-02	5	NB-F	7d	NI	Rio de Janeiro	5.75/5.79	93.8	34,979	MT933047
CHIKV-20	Serum	2019-07-31	11	F	46	NI	Rio de Janeiro	14.7	83.9	160,920	MT933048
CHIKV-21	Serum	2019-07-07	1	F	21	NI	Duque de Caxias	15.3	93.6	67,036	MT933049
CHIKV-22	Serum	2019-08-14	1	F	19	NI	Rio de Janeiro	14.8/15.5	92.7	76,999	MT933050
CHIKV-23	Serum	2019-05-28	2	M	63	Tijuca	Rio de Janeiro	17.32	93.5	58,496	MT933051

ID: Identification; F: Female; M: Male; NB: Newborn; m: month; d: day; NI: No information; Ct: cycle threshold.

**Table 2 pathogens-09-00984-t002:** Amino acid substitutions observed in the newly coding region of CHIKV from Rio de Janeiro, 2019 reported in this study.

			Polyprotein Region/Amino Acid (aa) Substitution																	
		Type of Protein	Nonstructural Protein (NSP)								Structural Protein (SP)									
		Protein	NSP2				NSP3		NSP4		E2				6k	E1				
		Sites (Protein)	57	352	452	466	545	334	111	238	74	178	248	377	52	98	151	211	269	305
**Strains**	**Reference**	**KP164568|BA|2014**	**A**	**P**	**V**	**M**	**A**	**A**	**I**	**W**	**T**	**R**	**L**	**V**	**M**	**A**	**D**	**K**	**V**	**A**
	**Clade A**	CHIKV-2	.	A	M	.	S	V	.	.	.	.	.	.	L	.	V	T	M	T
		CHIKV-3	.	A	.	.	S	.	.	C	.	.	.	.	.	.	V	T	M	T
		CHIKV-4	.	A	.	.	S	.	.	.	.	.	F	.	L	.	.	T	M	T
		CHIKV-5	.	A	.	L	S	.	.	.	.	.	.	.	L	.	.	T	M	T
		CHIKV-6	.	A	.	.	S	.	.	.	.	.	F	.	.	.	V	T	M	T
		CHIKV-7	.	A	.	.	S	.	.	.	M	.	F	.	L	.	V	T	M	T
		CHIKV-8	.	A	.	L	S	.	.	.	M	.	F	.	L	.	V	T	M	T
		CHIKV-9	.	A	.	.	S	.	.	C	M	.	F	.	L	T	V	T	M	T
		CHIKV-10	.	A	.	L	S	.	.	.	M	.	.	.	L	.	.	T	M	T
		CHIKV-12	.	A	.	.	S	.	.	C	M	.	F	.	.	.	V	T	M	T
		CHIKV-13	.	A	.	.	S	.	.	.	.	.	.	.	L	.	.	T	M	T
		CHIKV-14	.	A	M	.	S	V	.	.	.	.	.	I	L	.	V	T	M	T
		CHIKV-15	.	A	.	L	S	.	.	C	.	.	.	.	.	.	V	T	M	T
		CHIKV-16	.	A	.	L	S	.	.	C	.	.	.	I	.	.	V	T	M	T
		CHIKV-17	.	A	M	.	S	V	.	.	.	.	.	.	L	.	.	T	M	T
		CHIKV-18	.	A	.	.	S	.	.	.	.	.	.	.	L	.	.	T	M	T
		CHIKV-19	.	A	M	.	S	V	.	.	.	.	F	.	.	.	.	T	M	T
		CHIKV-20	.	A	.	.	S	.	.	C	M	.	F	I	L	T	V	T	M	T
		CHIKV-21	.	A	.	.	S	.	.	.	.	.	F	.	L	.	V	T	M	T
		CHIKV-22	.	A	.	.	S	.	.	.	M	.	F	I	L	T	V	T	M	T
		CHIKV-23	.	A	.	.	S	.	.	.	M	.	F	.	L	.	V	T	M	T
	**Clade B**	CHIKV-1	V	.	.	.	.	.	.	.	.	H	.	I	L	.	V	.	.	.
		CHIKV-11	V	.	.	.	.	.	.	.	M	H	F	.	L	T	V	.	M	.

Amino acid (aa) abbreviations: A: Alanine; V: Valine; P: Proline; M: Methionine; L: Leucine; R: Arginine; I: Isoleucine; W: Tryptophan; C: Cysteine; H: Histidine; F: Phenylalanine; D: Aspartic Acid; K: Lysine. Protein (amino acids numbers): Nonstructural protein (2474), NSP2 (798), NSP3 (524), NSP4 (611), Structural protein (1248), E2 (423), 6k (61), E1 (439).

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
