# Peer review of "Co-Circulation of Two Independent Clades and Persistence of CHIKV-ECSA Genotype during Epidemic Waves in Rio de Janeiro, Southeast Brazil"

_pathogens, 2020, doi:10.3390/pathogens9120984_

Round 1

Reviewer 1 Report

In this article, Allison Araújo Fabri et al discussed the introduction of two different strain of CHIKV- ECSA in RJ state and more interestingly they suggest the characterisation of nucleotide signature associated to conservative amino-acid. They also pointed out that up to now there is no ermergence of the mutation associated with increase replication of the CHIKV in albopictus or aegypti aedes mosquitoes.

The sequence screening/cleaning and phylogenetic analysis is really at the top of the current art.

My only major comment is about the discussion conclusion that might be to provide or suggest a molecular method to identify these two strains by a single PCR method or any system avoiding the need to do sequencing.

Whatever the conclusion, I have only minor comments 1) on the table 2 presentation in which the site position should be on a single column with a single digit on each line we can read from top to bottom for all position and not with two digits on two line that is slightly disturbing or worse with 2 digits in the first line and a single one on the following.

2) an Article from Sahadeo et al. deserved to be cited even that explore the Asian strain diversity and, a comparieson or comment about the evolution of the strains complex deserved to be done.

Understanding the evolution and spread of chikungunya virus in the Americas using complete genome sequences. Sahadeo NSD, Allicock OM, De Salazar PM, Auguste AJ, Widen S, Olowokure B, Gutierrez C, Valadere AM, Polson-Edwards K, Weaver SC, Carrington CVF. Virus Evol. 2017 May 3;3(1):vex010. doi: 10.1093/ve/vex010. eCollection 2017 Jan. PMID: 28480053 Free PMC article.

Author Response

Reviewer #1:

In this article, Allison Araújo Fabri et al discussed the introduction of two different strain of CHIKV- ECSA in RJ state and more interestingly they suggest the characterization of nucleotide signature associated to conservative amino-acid. They also pointed out that up to now there is no emergence of the mutation associated with increase replication of the CHIKV in albopictus or aegypti aedes mosquitoes.

We thank the reviewer for his positive comments

  • The sequence screening/cleaning and phylogenetic analysis is really at the top of the current art.

We thank the reviewer for his positive comments

  • My only major comment is about the discussion conclusion that might be to provide or suggest a molecular method to identify these two strains by a single PCR method or any system avoiding the need to do sequencing.

We thank the reviewer for his comment. As we state in our manuscript our phylogenetic reconstructions suggest that the 2019-CHIKV-ECSA epidemic in Rio de Janeiro state was characterized by the co-circulation of multiple clades, identified as clade A and B. Nevertheless, there is limited information about the genomic epidemiology of CHIKV circulating in Rio de Janeiro from genomic surveillance studies. The shortage of complete genomic sequences available impairs our understanding of the CHIKV introduction and establishment in the region. Thus, in this cryptic moment, we deeply believe that the priority need to be focus on generating more genomes sequences from the two distinct clade that posteriorly will allow to set up a single PCR method that will be able to quickly identify all the strain circulating and/or co-circulating in the region.

  • Whatever the conclusion, I have only minor comments 1) on the table 2 presentation in which the site position should be on a single column with a single digit on each line we can read from top to bottom for all position and not with two digits on two line that is slightly disturbing or worse with 2 digits in the first line and a single one on the following.

    We thank the reviewer for his comment. Changed as requested (new table 2).

  • An Article from Sahadeo et al. deserved to be cited even that explore the Asian strain diversity and, a comparison or comment about the evolution of the strains complex deserved to be done. (Understanding the evolution and spread of chikungunya virus in the Americas using complete genome sequences. Sahadeo NSD, Allicock OM, De Salazar PM, Auguste AJ, Widen S, Olowokure B, Gutierrez C, Valadere AM, Polson-Edwards K, Weaver SC, Carrington CVF. Virus Evol. 2017 May 3;3(1): vex010. doi: 10.1093/ve/vex010. eCollection 2017 Jan. PMID: 28480053 Free PMC article).

We agree with the reviewer’s suggestion and we have included the citation of the Shadeo et al. article in our manuscript.

Reviewer 2 Report

Overall a reasonably well conducted study looking at isolates in one part of Brazil.  Unfortunately, how this work fits into the grander scheme of CHIKV spread in South America is largely missing, so the context, significance, utility and implications are hard to understand.  For instance, no mention is made of introduction of ECSA into S. America generally.  Have other areas of the word seen the evolution of such distinct ECSA clades?  Also how does this relate to instruction of the Asia genotype?  Has the Asian genotype similarly seen to have been introduced multiple times into S. America?  Has the Asia genotype also adopted distinct clades in S. America?  The overall context perhaps with a map of America and all the purported introductions and associated dates would provide a framework and context for understanding the current data.  At the moment this is paper has a very narrow focus and thus not particularly interesting to a broader readership, who would want to understand how this affects our general understanding of how this virus has spread in S America. 

Perhaps the biggest omission is some context around the observation that the sequencing failed to see the A226V and K211E substitutions that were purported to be so critical to albopitus-mediated spread (Emerg Microbes Infect. 2020 Dec;9(1):1912-1918).  How widespread is this in S. America isolates? Is this due to preferential spread of ECSA by aegypti in S. America?  Does this cast further doubt on the role of these mutations in the global pandemic (Euro Surveill. 2018 May;23(22):1800246)?

The Abstract asserts CHIKV is associated with neurological complications, although this can be true, this is by no means the main clinical issue which is potentially debilitating polyarthralgia/polyarthritis (Nat Rev Rheumatol. 2019 Oct;15(10):597-611).

L190 How do these sequences assist public health measures?  I am unaware of any public health measures for CHIKV that rely on sequence data. The value here is to understand spread - hence the requirement to fit this data into a broader overall S. American context.

Minor issues

Not sure how introduction events can be “mediated” (l36 ) by the Northeastern region.  In what way is this region responsible and how is this related to spread in Brazil and S. America?

L119 events of CHIKV introduction – is this supposed to mean ECSA CHIKV introduction? Otherwise what about Asian genotype introduction? Similarly l156 – is this talking about Asian, ECSA or both?

L161 Not quite sure what this is corroborating;  “which corroborates with” also unclear English.  Two clases are found in RJ, two clades are found in Brazil, two clades are found globally?

Author Response

Reviewer #2:

Overall a reasonably well conducted study looking at isolates in one part of Brazil.  Unfortunately, how this work fits into the grander scheme of CHIKV spread in South America is largely missing, so the context, significance, utility and implications are hard to understand.  For instance, no mention is made of introduction of ECSA into S. America generally.  Have other areas of the word seen the evolution of such distinct ECSA clades?  Also how does this relate to instruction of the Asia genotype?  Has the Asian genotype similarly seen to have been introduced multiple times into S. America?  Has the Asia genotype also adopted distinct clades in S. America?  The overall context perhaps with a map of America and all the purported introductions and associated dates would provide a framework and context for understanding the current data.  At the moment this is paper has a very narrow focus and thus not particularly interesting to a broader readership, who would want to understand how this affects our general understanding of how this virus has spread in S America. 

We thank the reviewer for his comments. We agree with the reviewer that the challenging thing in this moment will be improve our understanding on the CHIKV transmission dynamics through the Americas, however prior to that there is the necessity to understand the complex pattern of CHIKV transmission between epidemic seasons and sampled locations also in a local scale in order to provide genomic data that could allow to identify the possible role that Brazil could have played as a source for national as well as international dispersion (of the CHIKV-ECSA genotype), enhanced by cross-border transmission also to other American countries, highlighting in this way the utility of combining genomic and evolutionary methods to understand ongoing mosquito-borne epidemics.

To the respect of the Asian genotype local transmission of this lineage was detected in Brazil for the first time in September 2014 in Oiapoque municipality, in the northern region, since then, only the ECSA genotype has been detected in several other Brazilian states in the north-eastern, south-eastern and northern regions, exerting a serious threat to public health, especially because the ECSA lineage is associated with higher symptomatic to asymptomatic ratio compared with the Asian lineage circulating in the Caribbean, this is why appear to be crucial provide more genomic data in order to assist in the monitoring and understanding of arbovirus epidemics, which might help to attenuate public health impact of infectious diseases.

To the respect of the co-circulation of distinct clades (identified as clade A and B in our reconstruction), we need to agree that this appear to be a pattern that have been already described also for other mosquito-borne viruses worldwide. Moreover, the shortage of complete genomic sequences available from the two clades impairs our understanding of the CHIKV introduction, re-introduction and establishment in this region. Thus, in this cryptic moment, we deeply believe that the priority need to be focus on generating more genomes sequences from the two distinct clade that posteriorly will also allow to set up a single PCR method that will be able to quickly identify all the strain circulating and/or co-circulating in the region.

  • Perhaps the biggest omission is some context around the observation that the sequencing failed to see the A226V and K211E substitutions that were purported to be so critical to albopictus-mediated spread (Emerg Microbes Infect. 2020 Dec;9(1):1912-1918).  How widespread is this in S. America isolates? Is this due to preferential spread of ECSA by aegypti in S. America?  Does this cast further doubt on the role of these mutations in the global pandemic (Euro Surveill. 2018 May;23(22):1800246)?

Studies have shown that several IOL strains presented a series of mutations including the E1-A226V and K211E adaptive mutation that confers an increased replication rate in Aedes albopictus, thus enhancing viral transmission potential (Schuffenecker I. et al., 2011; Tsetsarkin KA. et al., 2007; Vazeille M. et al., 2007). Moreover, by performing protein alignments of our obtained sequences we did not observe in our samples any of those mutations associated with increased CHIKV transmission in Ae. albopictus mosquitos. Thus, future investigation of the fitness, viral infectivity and evolution of CHIKV in mosquito populations, together with continued genomic surveillance, will determine whether the upsurge in the number of cases in Rio in 2019 was due to the acquisition of specific mutations that increase replication rates in local Aedes spp. mosquitoes. 

  • The Abstract asserts CHIKV is associated with neurological complications, although this can be true, this is by no means the main clinical issue which is potentially debilitating polyarthralgia/polyarthritis (Nat Rev Rheumatol. 2019 Oct;15(10):597-611).

We agree and we made the requested change to this line in the abstract, which now reads “Chikungunya virus infection in Brazil has raised several concerns due to the rapid dissemination of the virus and its association with several clinical complications”.

  • L190 How do these sequences assist public health measures?  I am unaware of any public health measures for CHIKV that rely on sequence data. The value here is to understand spread - hence the requirement to fit this data into a broader overall S. American context.

We thank the reviewer for his comment.

Genome sequencing has become a powerful tool for tracking virus transmission and this approach has been used in genomic epidemiology studies of arbovirus during epidemics in Brazil (Naveca FG et al., 2019; Faria et al., 2018; Faria et al., 2017; Giovanetti et al., 2020; Giovanetti et al., 2019). The analysis of full or near full length genomes maximize the usefulness of molecular clock models to infer time-scaled phylogenetic trees. Hence, by analysing heterochronous datasets with samples collected in different time points and/or locations, time-scaled phylogenies become a powerful tool to describe trends in epidemic spread (Shapiro B, et al., 2017). Thus, genomic data generated by real time portable sequencing technology can be employed and combined to epidemiological models in order to assist public health laboratories in monitoring and understanding the diversity of circulating mosquito-borne viruses.

Minor issues

  • Not sure how introduction events can be “mediated” (l36) by the Northeastern region.  In what way is this region responsible and how is this related to spread in Brazil and S. America?

We thank the reviewer for his comment.

Considering the state of art and the current knowledge we cannot provide evidence that the Brazilian Northeastern region could have play an important role in the dissemination of the CHIKV-ECSA genotype through the Americas, since more genomic data from Brazil and other Americas countries will be necessary. What we stated in our manuscript was that, our phylogenetic results suggested (with high posterior probability support) that since 2015, Rio de Janeiro has experienced two independent events of CHIKV introduction, both mediated by northeastern region where this lineage was first detected in late 2014 (as highlighted if figure 1) that appear to be the most likely source location of the ECSA-lineage strain that was circulating during the 2016-2019 epidemics in the state of Rio de Janeiro.

  • L119 events of CHIKV introduction – is this supposed to mean ECSA CHIKV introduction? Otherwise what about Asian genotype introduction? Similarly, l156 – is this talking about Asian, ECSA or both?

We thank the reviewer for his comment and we made the necessary changes to this line which now reads “Together, our results suggest that since 2015, Rio de Janeiro has experienced two independent events of CHIKV-ECSA introduction, both mediated by northeastern region where this lineage was first detected in late 2014”

  • L161 Not quite sure what this is corroborating; “which corroborates with” also unclear English.  Two classes are found in RJ, two clades are found in Brazil, two clades are found globally?

We thank the reviewer for his comment and we made the necessary changes to this line which now reads: “Our results revealed that the 2019-CHIKV-ECSA epidemic in RJ state was characterized by the co-circulation of two clades (clade A and B), in line with previous findings [13] highlighting that these two different clades are persistent and responsible for the latest epidemics registered in the state”.

To the respect of the identified clades, we are referring to the co-circulation of two distinct CHIKV-ECSA clades in the Rio de Janeiro state.

Reviewer 3 Report

In this study the authors look to understand how chikungunya virus (CHIKV) was introduced and has spread during the 2016-2019 outbreaks in Brazil. They obtained 23 clinical samples, sequenced the genomes, and performed a nice phylogenetic and SNV analysis of these samples. They conclude that there were two introductions of CHIKV in Rio de Janiero, forming two distinct clades of virus. They also identify novel mutations throughout the CHIKV genome. This study is well-executed and well-written. It provides novel insight into CHIKV emergence and add to the important of genomic surveillance. My concerns are minor.

1. I am a bit confused by the map and location of the study. First, for the map in Figure 1A, in the legend or figure, please tell us what the letters mean in the inset. SF? CF? etc. Second, in part C there appear to be only 2 areas where the samples are from. RJ and DC. What are these? Please add this to the figure legend. More importantly, from the table it looks like these are two cities (Maybe put the two cities on the map?) If this is the case, did you only sample from two cities and not from the whole state? I think it should be clarified in the text whether these samples were from all the regions of RJ state or two major areas. In particular, you conclude in line 175 that both introductions were from the Northeast region but that isn't clear to me with how things are labeled.

2. Table 2 is confusing with 3 reference genomes. It looks like you used the 2014 strain for alignments. What are the other two "references" for? I think you can remove the word reference from the left or remove them altogether if they don't add to the story.

Author Response

Reviewer #3:

In this study the authors look to understand how chikungunya virus (CHIKV) was introduced and has spread during the 2016-2019 outbreaks in Brazil. They obtained 23 clinical samples, sequenced the genomes, and performed a nice phylogenetic and SNV analysis of these samples. They conclude that there were two introductions of CHIKV in Rio de Janiero, forming two distinct clades of virus. They also identify novel mutations throughout the CHIKV genome. This study is well-executed and well-written. It provides novel insight into CHIKV emergence and add to the important of genomic surveillance. My concerns are minor.

We thank the reviewer for his positive comments.

  1. I am a bit confused by the map and location of the study. First, for the map in Figure 1A, in the legend or figure, please tell us what the letters mean in the inset. SF? CF? etc. Second, in part C there appear to be only 2 areas where the samples are from. RJ and DC. What are these? Please add this to the figure legend. More importantly, from the table it looks like these are two cities (Maybe put the two cities on the map?) If this is the case, did you only sample from two cities and not from the whole state? I think it should be clarified in the text whether these samples were from all the regions of RJ state or two major areas. In particular, you conclude in line 175 that both introductions were from the Northeast region but that isn't clear to me with how things are labeled.

We agree with the reviewer it looked really confuse. We have now made the necessary changes in the figure 1 and in the figure legend.

  1. Table 2 is confusing with 3 reference genomes. It looks like you used the 2014 strain for alignments. What are the other two "references" for? I think you can remove the word reference from the left or remove them altogether if they don't add to the story.

We totally agree with the reviewer. We have now made the necessary changes.

Round 2

Reviewer 2 Report

Not quite sure how to respond here as there has been almost no attempt to change the manuscript in line with the comments.  The responses to the review comments attempt to explain things to the reviewer but almost no attempts have been made to address the concerns in the manuscript.  The purpose of a review is surely to improve the manuscript, not to explain issues to the reviewer. 

The main concern remains - this paper has a very narrow focus with no attempts at placing the observations in some meaningful broader context.

Why the authors appear reluctant to describe the disease (e.g. rheumatic manifestations primarily presenting as acute and chronic polyarthralgia/polyarthritis) in the introduction (see suggested review) is perplexing; "several clinical complications" with no reference provided, certainly does not provide a useful description of the disease.

A geographic region cannot “mediate” viral spread.  The definition is “ bring about (a result such as a physiological effect).”  This is the wrong choice of verb. A geographic region can be the source of a new virus.

Author Response

Reviewer #2:

Not quite sure how to respond here as there has been almost no attempt to change the manuscript in line with the comments.  The responses to the review comments attempt to explain things to the reviewer but almost no attempts have been made to address the concerns in the manuscript. The purpose of a review is surely to improve the manuscript, not to explain issues to the reviewer. The main concern remains - this paper has a very narrow focus with no attempts at placing the observations in some meaningful broader context.

We thank the reviewer for his comments. We agree with the reviewer that our manuscript have a very specific topic and that the challenging thing in this moment would be instead try to get insight regarding the CHIKV transmission dynamics through the Americas, however prior to that there is the necessity to understand the complex pattern of CHIKV transmission between epidemic seasons and sampled locations also in a local scale in order to provide genomic data that could allow to identify the possible role that Brazil could have played as a source for national as well as international dispersion (of the CHIKV-ECSA genotype), enhanced by cross-border transmission also to other American countries, highlighting in this way the utility of combining genomic and evolutionary methods to understand ongoing mosquito-borne epidemics.

This is why we didn’t change the main focus of our manuscript, that was try to understand the 2019-ECSA-CHIKV epidemic in Rio de Janeiro, also because, despite such hyperendemicity since 2014, there is still relatively limited information on the evolution, transmission and spread of CHIKV within Brazilian states, which appear to be essential for epidemic preparedness, response and control. Combined to that there is still also, a huge paucity of complete genome sequences from this virus (just 150 complete genomes sequences from the Brazilian epidemics obtained over 8 years compared with the 113,642 SARS-CoV-2 genomes for example, obtained over just 9 months of epidemic), which really impairs our understanding on its transmission and dispersion through Brazil and the Americas. This is why we really believe that is crucial provide more genomic data from the most affected Brazilian regions and states, such as Rio de Janeiro, that in 2019 reported more than 86,264 CHIKV suspected cases until epidemiological week 52, which is approximately 65% of all the probable cases notified in the country.

  1. Why the authors appear reluctant to describe the disease (e.g. rheumatic manifestations primarily presenting as acute and chronic polyarthralgia/polyarthritis) in the introduction (see suggested review) is perplexing; "several clinical complications" with no reference provided, certainly does not provide a useful description of the disease.

We thank the reviewer for his comments. Considering that the abstract has some limitation regarding the number of characters we make just few editing in the abstract section considering the comments the reviewer provides us during the first round of revisions, but we provided more details regarding the diseases in the introduction section, which now reads:

“Chikungunya virus (CHIKV) is an RNA alphavirus belonging to the Togaviridae family, transmitted by Aedes aegypti and by A. albopictus mosquitoes [1,2]”. Infection with CHIKV typically causes a self-limiting febrile illness, chikungunya fever and common clinical manifestations of the disease include fever, muscle pain, rash and severe joint pain, which may last for months to years [1]. Chikungunya fever appear also to be linked with long-lasting rheumatic disorders presenting as acute and chronic polyarthralgia/polyarthritis, that leads to functional impairment affecting daily living activities up to several years after infection [1, 2].

Round 3

Reviewer 2 Report

I have now seen this manuscript 3 times and the authours seem to have largely ignored the suggestions to improve the manuscript; eg no context, very dated references  etc

Author Response

I have now seen this manuscript 3 times and the authours seem to have largely ignored the suggestions to improve the manuscript; eg no context, very dated references etc.

We would like to thank you the reviewer for all his suggestions that muchly have improved our manuscript. Since the first round of revision we have considered all the reviewer recommendations. Since than we have made the required changes in the abstract, introduction sections, in the table and now as suggested in the references used. One of the main points of the reviewer was the focus of our research article, that was try to understand the 2019-ECSA-CHIKV epidemic in Rio de Janeiro. We didn’t change it because we strongly believe that, despite such hyperendemicity since 2014, there is still relatively limited information on the evolution, transmission and spread of CHIKV within Brazilian states, which appear to be essential for epidemic preparedness, response and control. Combined to that there is still also, a huge paucity of complete genome sequences from this virus (just 150 complete genomes sequences from the Brazilian epidemics obtained over 8 years compared with the 113,642 SARS-CoV-2 genomes for example, obtained over just 9 months of epidemic), which really impairs our understanding on its transmission and dispersion through Brazil and the Americas. This is why we really believe that is crucial provide more genomic data from the most affected Brazilian regions and states, such as Rio de Janeiro that in 2019 reported more than 86,264 CHIKV suspected cases until epidemiological week 52, which is approximately 65% of all the probable cases notified in the country.